# β-aminopropionitrile-induced thoracic aortopathy is refractory to cilostazol and sildenafil in mice

Samuel C. Tyagi[1,2,3,4], Sohei Ito[1,2], Jacob C. Hubbuch[1,2,3,4], Michael K. Franklin[1,2], Deborah A. Howatt[1,2], Hong S. Lu[1,2,3], Alan Daugherty[1,2,3*], Hisashi Sawada[1,2,3*]

1 Saha Cardiovascular Research Center, College of Medicine, University of Kentucky, Lexington, Kentucky, United States of America, 2 Saha Aortic Center, College of Medicine, University of Kentucky, Lexington, Kentucky, United States of America, 3 Department of Physiology, College of Medicine, University of Kentucky, Lexington, Kentucky, United States of America, 4 Department of Surgery, College of Medicine, University of Kentucky, Lexington, Kentucky, United States of America

* hisashi.sawada@uky.edu (HS); Alan.Daugherty@uky.edu (AD)

## Abstract

Thoracic aortopathies are life-threatening diseases including aneurysm, dissection, and rupture. Cilostazol, a phosphodiesterase (PDE) 3 inhibitor, and sildenafil, a PDE5 inhibitor, have been used clinically for peripheral arterial disease and erectile dysfunction or pulmonary hypertension, respectively. Recent studies report their effects on abdominal aortic aneurysm formation. However, their impacts on thoracic aortopathy remain unknown. In this study, we investigated whether cilostazol and sildenafil affect thoracic aortopathy induced by β-aminopropionitrile (BAPN) administration in mice. Bulk RNA sequencing analysis revealed that BAPN administration upregulated *Pde3a* transcription in the ascending aorta and *Pde5a* in both ascending and descending regions before thoracic aortopathy formation. Next, we tested the effects of cilostazol or sildenafil on BAPN-induced thoracic aortopathy. BAPN-administered mice were fed a diet supplemented with either cilostazol or sildenafil. Mass spectrometry measurements determined the presence of cilostazol or sildenafil in the plasma of mice fed drug-supplemented diets. However, neither drug altered BAPN-induced aortic rupture nor aneurysm formation and progression. These results provide evidence that cilostazol and sildenafil did not influence BAPN-induced thoracic aortopathy in mice.

## Introduction

Thoracic aortopathy is a spectrum of diseases, including aneurysm, dissection, and rupture. A key pathological feature of thoracic aortopathy is extracellular matrix (ECM) remodeling, including elastic fiber fragmentation and excessive collagen fiber deposition [1]. Given the vital role of ECM in maintaining structural integrity, preventing ECM changes may inhibit development and progression of thoracic aortopathy. Among available mouse models, β-aminopropionitrile (BAPN) administration

**Data availability statement:** All relevant data are within the paper and its Supporting information files.

**Funding:** The studies reported in this article were supported by the National Heart, Lung, and Blood Institute of the National Institutes of Health (R35HL155649), the American Heart Association (23MERIT1036341, 24CDA1268148), and the Leducq Foundation for the Networks of Excellence Program (Cellular and Molecular Drivers of Acute Aortic Dissections).

**Competing interests:** The authors have declared that no competing interests exist.

is increasingly used in preclinical studies [2–6]. BAPN inhibits elastin and collagen crosslinking by blocking lysyl oxidase activity, leading to ECM changes that are a ubiquitous feature of thoracic aortopathy. Therefore, BAPN administration may be an approach to investigate mechanisms and therapeutic potential of thoracic aortopathy.

Phosphodiesterases (PDEs) regulate intracellular cyclic nucleotide abundance by catalyzing their hydrolysis into inactive forms [7,8]. Within the PDE family, PDE3 and PDE5 have garnered interest in their potential roles in the pathophysiology of aortic diseases [9–13]. Cilostazol, a PDE3 inhibitor used clinically for peripheral arterial disease, has vasodilatory and antiplatelet effects [14]. A recent preclinical study demonstrated that cilostazol attenuated angiotensin II-induced abdominal aortic aneurysms in hypercholesterolemic mice [9]. In contrast, sildenafil, a PDE5 inhibitor used for erectile dysfunction and pulmonary hypertension, exacerbated abdominal aortic aneurysm induced by a combination of BAPN oral administration and elastase topical application in normolipidemic mice [13]. These findings provide evidence that cilostazol and sildenafil have differential effects on abdominal aortic aneurysms. However, it remains unknown whether these drugs affect thoracic aortopathy formation and progression. The mechanisms underlying aortopathy differ between the thoracic and abdominal regions [15], highlighting a need for preclinical studies to determine the effects of these drugs on thoracic aortopathy.

Based on clinical and preclinical evidence, we hypothesized that cilostazol attenuates thoracic aortopathy, whereas sildenafil exacerbates it. In this study, we examined the effects of cilostazol and sildenafil on thoracic aortopathy associated with changes in ECM. BAPN-administered mice were fed a standard laboratory diet or a diet supplemented with cilostazol or sildenafil, and thoracic aortopathy formation and progression were compared across groups.

## Materials and methods

### Mice

C57BL/6J mice at 3 weeks of age were purchased from The Jackson Laboratory (#000664) and housed in individually ventilated cages (5 mice/cage) under a 14-hour light and 10-hour dark cycle. Teklad Sani-Chip bedding (#7090A, Inotiv) was used for cage bedding. Mice were fed a standard rodent laboratory diet (#2918, Inotiv), cilostazol-supplemented diet (0.315 g/kg, #TD.190797, Teklad), or sildenafil-supplemented diet (0.315 g/kg, #TD.190796, Teklad) (Fig 1). Five days after initiating dietary interventions, BAPN (0.5% wt/vol, #A3134-25G, Millipore-Sigma or #A0796-500G, TCI Co.) was administered in drinking water. Our previous study revealed that BAPN administration for 28 days induced acute aortic pathologies, and 84-day administration results in progressive vascular remodeling. Therefore, BAPN was administered for 28 and 84 days to evaluate the effects of drugs on both the initiation and progression of thoracic aortopathy, respectively [6]. BAPN-induced thoracic aortopathy does not display significant sexual dimorphism [6]. Therefore, the present study evaluated the effects of PDE inhibition in male mice. Based on our previous study [6], the present study was designed with a sample size of n = 15 per group to achieve a statistical power greater than 0.8 to detect a 1.1 mm difference

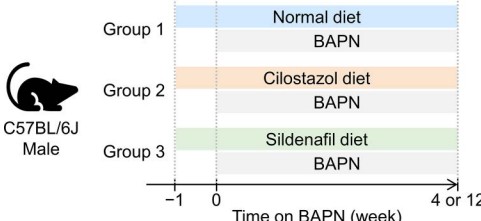

**Fig 1. Scheme for experimental designs.** BAPN was administered to male C57BL/6J mice fed a normal, cilostazol-, or sildenafil-supplemented diet for either 4 or 12 weeks.

with a standard deviation of 0.9, assuming an alpha level of 0.05. Mice were randomly assigned to the study groups, and no mice were excluded from the study. Drinking water was replaced twice weekly. All animal experiments adhered to the ARRIVE (Animal Research: Reporting of In Vivo Experiments) guidelines and were approved by the University of Kentucky Institutional Animal Care and Use Committee (2018–2967).

## Determination of aortic rupture and dimensions

All study mice were monitored at least once daily. Necropsies were performed immediately upon discovery of deceased mice to determine the cause of death. Aortic rupture was defined by the presence of extravascular blood accumulated in the thoracic or abdominal cavity. The site of blood egress was identified based on the location of the blood clot and a discernible disruption of the aortic wall. Mice were euthanized via intraperitoneal injection of ketamine (90 mg/kg, #11695-6840-1, Covetrus) and xylazine (10 mg/kg, #11695-4024-1, Covetrus) cocktail at 28 or 84 days following BAPN administration. The right atrial appendage was excised, and saline (~10 mL) was perfused via the left ventricle. After removing periaortic tissues, a black plastic sheet was inserted underneath the aorta [5,16]. A ruler was placed adjacent to the aorta for measurement calibration. Then, in situ aortic images were captured using a stereoscope (SMZ25, Nikon). Aortic diameters were measured perpendicularly to the aortic axis at the most dilated area of ascending and descending regions using NIS-Elements AR software (v5.11, Nikon). Measurements were verified by an individual who was blinded to the identity of the study groups.

## Mass spectrometry

Plasma cilostazol and sildenafil concentrations were measured using mass spectrometry in the Research Mass Spectrometry and Proteomics Core at the University of Kentucky. Sildenafil (1 mg/mL in solvent, Cerilliant S-010), cilostazol (powder, Sigma PHR1503-1G), and d8-sildenafil (1 mg/mL, LGC TRC-5435003) standards were used. Solvents included LC-MS grade methanol (Millipore MX0486), methyl tert-butyl ether (MTBE, Fisher E127), LCMS-Water, formic acid, and ammonium formate. Drug-free serum was used as the biological matrix. Chromatographic separation was performed on an ACQUITY UPLC BEH C18 column [130Å, 50 mm x 2.1 mm, 1.7 μm particle size] (Waters Part# 186002350).

Sildenafil and cilostazol were extracted from serum via liquid-liquid extraction. Briefly, serum (100 μL) was spiked with appropriate standards or methanol (for negative controls). Internal standard (d8-sildenafil, 100 μL at 80 ng/mL) was added to all samples. Samples were extracted with methyl tert-butyl ether (MTBE; 5 mL) by roto-racking for 5 minutes, and centrifugation at 2,500 rpm (~1000 x g) for 5 minutes. The MTBE layer was transferred to a clean tube and evaporated to dryness under nitrogen at 40°C. Analytes were reconstituted in 100 μL of mobile phase A (85% vol/vol) and mobile phase B (15% vol/vol). Mobile phase A was 5 mM ammonium formate in formic acid (0.01% vol/vol) in water, and mobile phase B was 100% methanol. Resuspended samples were transferred to silanized vials for LC-MS/MS analysis. Calibrators were

prepared by adding sildenafil and cilostazol to drug-free serum to achieve concentrations ranging from 1 to 200 ng/ml for each analyte. Quality control samples contained sildenafil and cilostazol at 60 ng/mL, respectively.

Extracted samples were analyzed on a Thermo Scientific TSQ Altis Plus Triple Quadrupole Mass Spectrometer coupled to a Vanquish Flex liquid chromatography system. Chromatographic separation was performed on an ACQUITY UPLC BEH C18 column (50 mm x 2.1 mm, 1.7 μm particle size; Waters). Mobile phases were: A, 5 mM ammonium formate in formic acid (0.1% vol/vol) in water; and B, methanol (100% vol/vol). Compounds were separated across a linear gradient from 1% mobile phase B to 100% mobile phase B over 5 minutes. The flow rate was set to 0.5 ml/min. The column temperature was set to 40°C. Detection was performed via multiple reaction monitoring (MRM) in positive electrospray ionization mode. Instrument parameters were set to the following: positive spray voltage 3000 V, sheath gas: 50, aux gas: 1, sweep gas: 1, ion transfer tube temperature: 325°C, vaporizer temperature: 350°C, Q1 resolution was set to 0.7 (FWHM), Q3 resolution was set to 1.2 (FWHM), CID gas (mTorr): 1.5.

### RNA sequencing data analysis

Read count data were obtained from our previous data posted on the Genome Expression Omnibus (GSE241968, RRID:SCR_005012) [6] and normalized using the "trimmed mean of M values" method in edgeR (v3.36.0, RRID:SCR_012802) on R (v4.1.0). Subsequently, normalized read counts for *Pde3a* and *Pde5a* were extracted. In the sequencing, aortic samples were harvested from ascending and descending regions at 1 week of BAPN administration, corresponding to the pre-pathological phase of thoracic aortopathy. Read count data of *Pde3a* and *Pde5a* were extracted and compared between vehicle- and BAPN-administered mice in ascending and descending regions. The R code used in this study is available upon request.

### Statistical analyses

Data are represented as individual data points with the median and 25th/75th percentiles. TMM-normalized read count data were analyzed by Student's t-test in ascending and descending regions separately. Log-Rank test was used to compare survival rates between groups in Kaplan-Meier curves. Normality and homogeneity of variance were assessed by Shapiro-Wilk and Brown-Forsythe tests, respectively. One-way analysis of variance followed by Holm-Sidak test was performed for parametric comparisons, whereas Kruskal-Wallis followed by Dunn's post-hoc test was performed for non-parametric comparisons. $P < 0.05$ was considered statistically significant. Statistical analyses were performed using SigmaPlot version 15.0 (SYSTAT Software Inc., RRID:SCR_003210).

## Results

### *Pde3a* and *Pde5a* mRNA were upregulated in the pre-pathological phase of the thoracic aorta in BAPN-administered mice

We first investigated the transcriptomic changes of aortic PDE3 and PDE5 in BAPN-administered mice using our previous bulk RNA-sequencing data [6]. While no statistical difference was observed in the descending region, *Pde3a* mRNA abundance was increased significantly in the ascending aorta of BAPN-administered mice compared to vehicle-administered mice (Fig 2A). *Pde5a* mRNA abundance was increased in both regions of BAPN-administered mice (Fig 2B). These data demonstrated upregulation of aortic PDE3 and PDE5 in the prepathological phase of BAPN-administered mice.

### Verification of cilostazol and sildenafil administration in mice fed drug-supplemented diet

To determine the role of PDE3 and PDE5 in development of BAPN-induced thoracic aortopathy, BAPN-administered mice were fed a diet supplemented with either cilostazol or sildenafil. A standard laboratory diet was administered to the BAPN-administered mice as a control group (Fig 1). After 28 days of BAPN administration, mice were euthanized. Mass

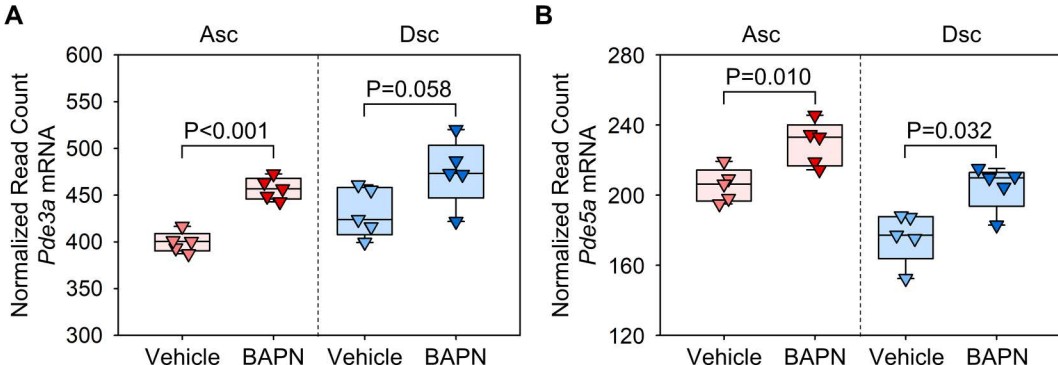

**Fig 2. Increase of *Pde3a* and *Pde5a* mRNA abundance in the thoracic aorta of BAPN-administered mice.** Normalized read count data from bulk RNA sequence data (GSE241968) for (A) *Pde3* and (B) *Pde5* mRNA in ascending (Asc) and descending (Dsc) aortas of mice administered with either vehicle or BAPN for 1 week. P values were determined by Student's t-test (A) or Mann-Whitney test (B) in each region.

spectrometry analysis was performed on plasma samples of these mice as well as those on a standard diet to validate the presence of these drugs in plasma. As expected, neither cilostazol nor sildenafil was detected in the plasma of mice fed a standard laboratory diet (Fig 3A, B). The cilostazol-supplemented diet increased plasma cilostazol concentrations, while the sildenafil-supplemented diet increased only sildenafil concentrations (Fig 3A, B). These data indicate the effective delivery of drugs using diet supplementation in BAPN-administered mice.

### Cilostazol and sildenafil did not affect BAPN-induced thoracic aortopathy formation

Subsequently, aortic phenotypes were evaluated after 28 days of BAPN administration. Consistent with previous reports [6], BAPN (0.5% wt/vol) administration resulted in high mortality due to aortic rupture and dissection in C57BL/6J mice. Aortic rupture-related death was observed in 44% of control mice during BAPN administration (Fig 4A). Although the mass spectrometry analysis validated the presence of cilostazol and sildenafil in the plasma (Fig 3A, B), cilostazol or sildenafil supplementation did not change the survival rate compared to mice fed control diet (Fig 4A). Maximum thoracic aortic

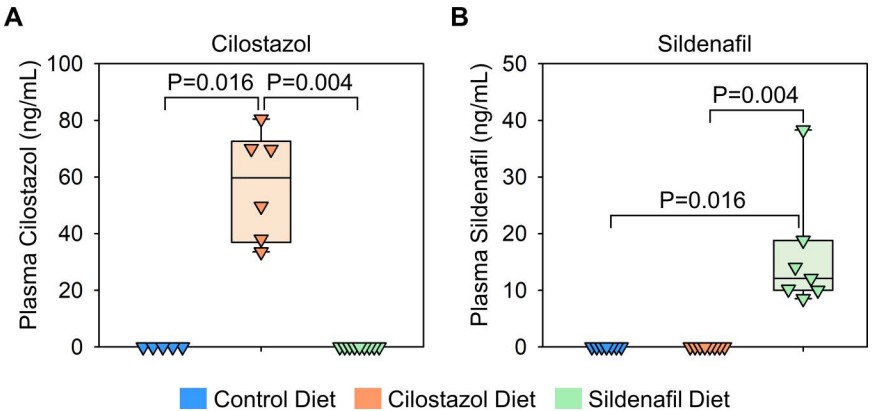

**Fig 3. Determination of plasma concentrations of cilostazol and sildenafil in mice fed drug-supplemented diet.** Plasma concentrations of (A) cilostazol and (B) sildenafil in mice fed control, cilostazol-, or sildenafil-contained diet. N = 5-9/group. P-values were determined by Kruskal-Wallis test followed by Dunn's test.

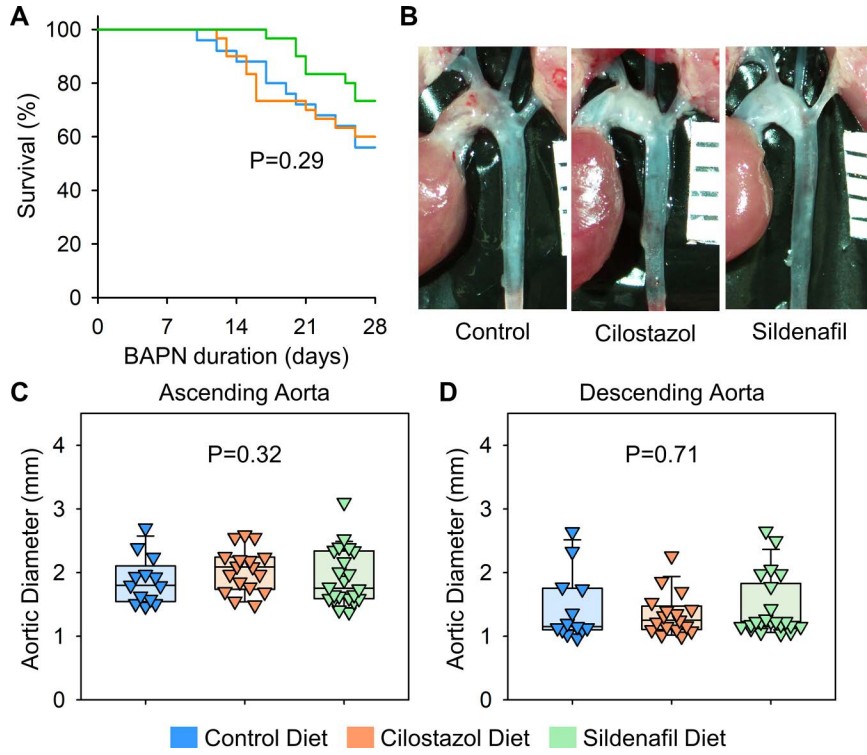

**Fig 4. Cilostazol or sildenafil did not change BAPN-induced thoracic aortopathy after 28 days of administration.** (A) Survival rate, (B) gross appearance of thoracic aortas, and aortic diameters in (C) ascending and (D) descending regions of control, cilostazol, or sildenafil-administered mice. P values were calculated by Log-Rank for (A) or Kruskal Wallis tests for (C, D).

diameters, measured using in situ aortic images, were comparable among three groups regardless of aortic regions (Fig 4B–D).

## Cilostazol and sildenafil did not affect BAPN-induced thoracic aortopathy progression

We then extended the study duration to 12 weeks to examine the effects of cilostazol and sildenafil on the progression of BAPN-induced thoracic aortopathy. Prolonged BAPN administration resulted in 68% mortality due to aortic rupture or dissection (Fig 5A). Similar to findings with 4 weeks of BAPN administration, cilostazol- and sildenafil-supplemented diets did not affect survival rates during 12 weeks of BAPN administration (Fig 5A). In situ aortic measurements showed that neither diet prevented nor exacerbated BAPN-induced thoracic aneurysm formation in the ascending or descending aorta after 12 weeks of BAPN administration (Fig 5B–D).

## Discussion

In the present study, we found that aortic *Pde3a* transcription in the ascending region and *Pde5a* transcription in both ascending and descending regions were upregulated before thoracic aortopathy formation during BAPN administration. Although the presence of cilostazol and sildenafil was detected, neither of the two drugs changed the development and progression of BAPN-induced thoracic aortopathy in mice.

A study reported that cilostazol attenuated angiotensin II-induced abdominal aortic aneurysm in hypercholesterolemic mice [9]. The protective effects of cilostazol were associated with the suppression of inflammatory cytokine expression. Since vascular inflammation is involved in the pathophysiology of BAPN-induced thoracic aortopathy [17], we

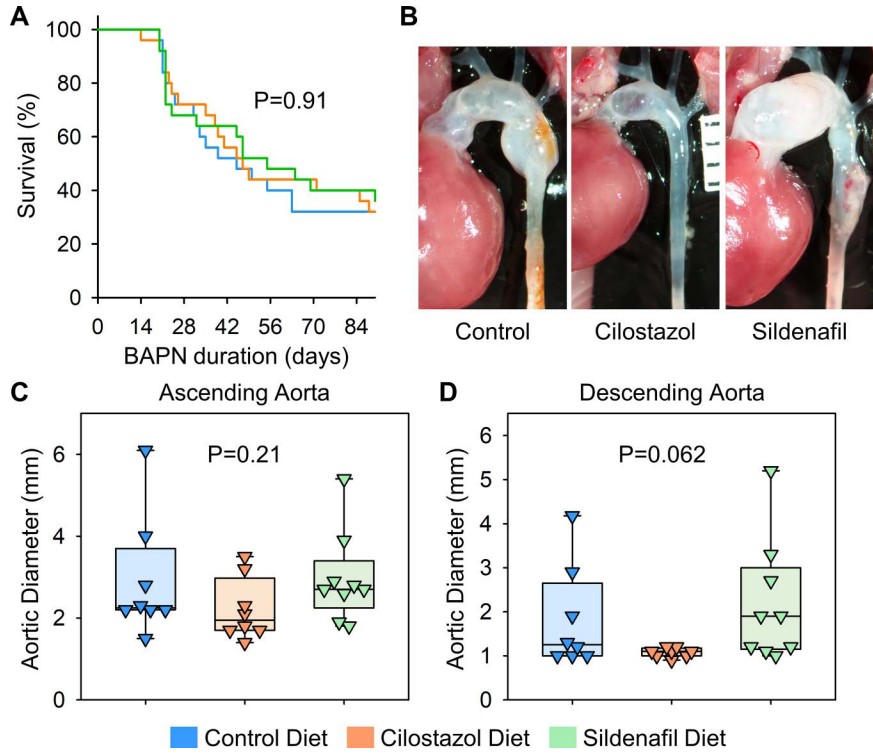

**Fig 5. Cilostazol or sildenafil did not suppress the progression of BAPN-induced thoracic aortopathy after 84 days of administration.**
(A) Survival rate, (B) gross appearance of thoracic aortas, and aortic diameters in (C) ascending and (D) descending regions of control, cilostazol, or sildenafil-administered mice. P values were calculated by Log-Rank for (A) or Kruskal Wallis tests for (C, D).

hypothesized that cilostazol similarly inhibits disease progression in this model. However, in the present study, cilostazol failed to suppress the formation and progression of thoracic aortopathy in BAPN-administered mice. BAPN-induced thoracic aortopathy results from the direct inhibition of lysyl oxidase-mediated ECM maturation, leading to structural destabilization of the aortic wall [6]. Therefore, inflammatory responses in this model appear to be a secondary consequence rather than a primary driver of disease development, which may explain the ineffectiveness of cilostazol in suppressing BAPN-induced thoracic aortopathy.

Sildenafil has been clinically used to treat erectile dysfunction and pulmonary hypertension via its vasodilatory effect through the increase of nitroxide production [18–20]. A previous study found that sildenafil enhanced progression of abdominal aortic aneurysms induced by BAPN administration and topical elastase application [13]. This effect was attributed to impaired aortic contractility, characterized by increased protein kinase G abundance and decreased myosin light chain phosphorylation [13]. However, in our study, sildenafil did not affect the formation and progression of BAPN-induced thoracic aortopathy. Although both studies used BAPN to induce aortopathy, the disease locations differ. There is evidence that aortopathy exhibits regional heterogeneity that are attributable to distinct disease mechanisms [15]. This indicates that BAPN-induced thoracic aortopathy may not be driven by protein kinase G activation and myosin light chain phosphorylation. Moreover, bulk RNA sequencing in our previous study demonstrated that the molecular mechanisms driving BAPN-induced thoracic aortopathy are independent of the transforming growth factor β signaling pathway and the renin-angiotensin system [6], both of which are commonly implicated in aortopathy mechanisms. These results indicate that BAPN-induced thoracic aortopathy is mediated by mechanisms primarily driven by extracellular matrix disruption, distinguishing it from other aortopathy models. Further studies are required to elucidate the precise mechanisms

underlying BAPN-induced thoracic aortopathy. In particular, comprehensive histological analyses will be essential to identify the underlying cellular and molecular changes. In addition, single-cell RNA sequencing may provide deeper insights into downstream pathway mechanisms.

Although neither drug affected BAPN-induced thoracic aortopathy formation and progression, measurements of drug concentrations using mass spectrometry confirmed that dietary supplementation with cilostazol and sildenafil increased their plasma concentrations in mice in this study. Throughout the study, mice had continuous access to drug-supplemented diets. However, plasma half-lives of cilostazol and sildenafil in rodents are relatively short (cilostazol: 2–4 hours, sildenafil: 0.3–2 hours, in rats) [21,22]. Therefore, it remains unclear whether their plasma concentrations remained consistently elevated throughout the experiment. In addition, we used a drug concentration of 0.315 g/kg in the food, which corresponds to an estimated drug intake of approximately 1.3 mg/day, based on the commonly reported feeding behavior of C57BL/6J mice (4 g food/day) [23,24]. Previous studies investigating the effects of these drugs in several types of cardiovascular and cerebral diseases, such as heart failure and cognitive dysfunction [25–31]. While many of these studies using sildenafil employed doses around 1.3 mg/day, some studies investigating cilostazol administered higher doses of up to 60 mg/day. Therefore, it remains unknown whether higher doses can attenuate thoracic aortopathy. it remains unknown whether higher doses can attenuate thoracic aortopathy. Future studies are needed to evaluate the effects of continuous administration and higher doses of these drugs on BAPN-induced thoracic aortopathy formation and progression. Also, frequent plasma sampling and assessment of downstream signaling pathways, such as cyclic adenosine or guanosine monophosphates, are important to validate effective cilostazol and sildenafil administrations.

In the present study, BAPN (0.5% wt/vol in drinking water) was used to induce thoracic aortopathy in young male C57BL/6J mice. Importantly, our previous study demonstrated that BAPN concentrations in drinking water is a critical determinant of disease development and severity in this model [6]. While 0.1% BAPN led to minimal pathology and no mortality, increasing the concentration to 0.3% resulted in a significant incidence of aortic rupture (60%), with surviving mice developing thoracic aortopathy. The highest BAPN concentration used (0.5% wt/vol) concentration was even more severe, with an 80% mortality rate due to rupture, and the surviving mice exhibited extensive aortic pathology, including profound dilatation and involvement of the descending thoracic aorta. These findings highlight the dose-dependent effects of BAPN-induced thoracic aortopathy. Based on our previous studies examining the dose-response of BAPN [6], the present study used 0.5% BAPN to ensure a sufficiently high disease incidence that would allow robust assessment of phenotypic outcomes. We acknowledge that this high concentration leads to higher mortality, potentially interfering with data interpretation for the sildenafil group, since we hypothesized that inhibiting PDE5 augments thoracic aortopathy.

In conclusion, this study provides evidence that cilostazol and sildenafil do not influence BAPN-induced thoracic aortopathy in mice.

## Supporting information

**S1 File. Supplement excel file.**
(XLSX)

## Author contributions

**Data curation:** Samuel C. Tyagi, Sohei Ito, Jacob C. Hubbuch, Michael K. Franklin, Deborah A. Howatt.

**Formal analysis:** Samuel C. Tyagi, Hisashi Sawada.

**Funding acquisition:** Alan Daugherty, Hisashi Sawada.

**Investigation:** Samuel C. Tyagi.

**Methodology:** Samuel C. Tyagi.

**Supervision:** Hong S. Lu, Alan Daugherty, Hisashi Sawada.

**Validation:** Hong S. Lu, Alan Daugherty, Hisashi Sawada.

**Visualization:** Hisashi Sawada.

**Writing – original draft:** Samuel C. Tyagi, Hisashi Sawada.

**Writing – review & editing:** Samuel C. Tyagi, Sohei Ito, Hong S. Lu, Alan Daugherty.

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
