## [Decision Letter · Decision Letter 0]

9 May 2025

PONE-D-25-15324β-aminopropionitrile-induced thoracic aortopathy is refractory to cilostazol and sildenafil in micePLOS ONE

Dear Dr. Sawada,

Thank you for submitting your manuscript to PLOS ONE. After careful consideration, we feel that it has merit but does not fully meet PLOS ONE’s publication criteria as it currently stands. Therefore, we invite you to submit a revised version of the manuscript that addresses the points raised during the review process.

We look forward to receiving your revised manuscript.

Kind regards,

Jeffrey S Isenberg, MD, MPH

Academic Editor

PLOS ONE

2. To comply with PLOS ONE submissions requirements, in your Methods section, please provide additional information regarding the experiments involving animals and ensure you have included details on methods of sacrifice.  

 [The studies reported in this article were supported by the National Heart, Lung, and Blood Institute of the National Institutes of Health (R35HL155649), the American Heart Association (23MERIT1036341, 24CDA1268148), and the Leducq Foundation for the Networks of Excellence Program (Cellular and Molecular Drivers of Acute Aortic Dissections).]. 

Additional Editor Comments:

The reviewers and associate editor found manuscript interesting but substantial comments should be carefully reviewed. A response will require new data.

Reviewers' comments:

Reviewer's Responses to Questions

**Comments to the Author**

1. Is the manuscript technically sound, and do the data support the conclusions?

Reviewer #1: Partly

Reviewer #2: Yes

2. Has the statistical analysis been performed appropriately and rigorously? 

Reviewer #1: I Don't Know

Reviewer #2: Yes

3. Have the authors made all data underlying the findings in their manuscript fully available?

Reviewer #1: Yes

Reviewer #2: Yes

4. Is the manuscript presented in an intelligible fashion and written in standard English?

Reviewer #1: Yes

Reviewer #2: Yes

5. Review Comments to the Author

Reviewer #1: In the manuscript titled “β-aminopropionitrile-induced thoracic aortopathy is refractory to cilostazol and sildenafil in mice” the authors test the notion that altering the secondary degradation of cGMP by limiting the PDEs that do this would alert the outcome of vascular degradation in the aorta in mice given β-aminopropionitrile. This agent prevents stability in the elastic fibers that are essential to major artery function and mechanical health. It is a chemical model that produces in short order what occurs with time in some people, namely loss of functional elastic fibers in the aorta. PDE blockers have been in use for cardiovascular issues for years. They target several members of a large class of agents that are a second line of control of essential cGMP, the latter being the major signal agent of vascular nitric oxide generated by endothelial cells. Mice were given two agents, cilostazol or sildenafil in the diet and levels tacked with mass spec. Neither seemed to decrease aortic aneurysms rates.

A version of this manuscript was published on-line as a pre-print: bioRxiv [Preprint]. 2025 Mar 27:2025.03.24.645113.

Novelty: Some work has been done with β-aminopropionitrile in rodents. A PubMed search with the term “β-aminopropionitrile aorta mice” yielded ~130 papers. Indeed, some of this work is from the authors. For example, Arterioscler Thromb Vasc Biol. 2024 Jul;44(7):1555-1569. Of note this latter paper provides details about the murine model. But this raises the question, why the treatment was not applied to mice when they were phenotype analysis for the 2024 paper. While data in both papers seems helpful, I do not understand why the authors did not combine the work into a single paper.

Rationale: The Introduction offers a rationale based on data showing conflicting effects of the two treatments with one limiting aneurysm formation and the other worsening them. Thus, it seems reasonable to consider what accounts for this. However, in truth the models of arterial disease were not the same so that alone might be a reason for using one model and testing both agents.

Methods: The text on the timing of treatments and injury is a little confusing. Also, why give β-aminopropionitrile for 28 or 84 days? What was the reason for looking at plasma levels of the treatments; perhaps as a means to estimate the effective delivered amounts? Did you suspect that mice were not taking the agents by eating less. How many samples over time were taken and how did this compare to the administration of the treatments. I would be surprised that half-lives of these agents are not published for mice and rats. Provide any available data on food and water intake, activity, and weight changes during the study intervals. Justify using just male mice. Provide a power analysis for cohort sizes. Provide any vitals and cardiovascular information on the mice -BP, pulse, cardiac output, ejection fraction, etc. if available. If not give us reasons why this was not checked as a method to ascertain if the agents made any difference globally, and not just at the aorta.

I am concerned about the study design. As the authors published already on the high mortality of the stress (>40%) why go out to the extent that the same was achieved in the present study. This would seem to be unnecessary as one could look at early time points to find if the treatments made any difference.

Data: The whole tissue mRNA data is not too helpful. What should be done is single cell seq followed if indicated with bulk analysis. We just do not what to make of increase PDE mRNA - is it form immune cells, fibroblasts, smooth muscle cells, endothelial cells. They all respond to NO and cGMP and so have counter controls like PDEs. I suggest trying to localize this information. Maybe high contrast and magnification immunofluorescence of the aortic walls would help.

How did you validate the morphometric approaches used on the resected arch and descending aorta? I think in vivo doppler or other non-invasive approaches would be much more helpful here.

The paper has a limit in mechanism. To begin why not map the NO-cGMP-PKG pathway in the samples from the mice; the target might be PKG. As well, while transcripts to several PDEs changed there is no explanation as to why? Was it feedback from a lock of NO-cGMP signal. And what would account for the loss of NO-cGMP or any relative resistance to the same? Did the β-aminopropionitrile cause secondary/tertiary modifications on pathway molecules that altered their function? And/or did it upregulate known inhibitors of the NO-cGMP-PKG cascade such as thrombopsindin-1, superoxide, etc. (see: Proc Natl Acad Sci U S A. 2005 Sep 13;102(37):13141-6 and Nat Rev Cancer. 2009 Mar;9(3):182-94). I would bet a link to β-aminopropionitrile and superoxide is hinted at but not fully defined, while low NO turns on thrombospondin-1 expression (see Proc Natl Acad Sci U S A. 2005 Sep 13;102(37):13147-52).

Legends: Add the exact statistics test used for each graph.

Figures: Ok but was all of the color needed?

Indicate if AI was or was not used in manuscript perpetration and a section on what each author contributed. As well, mention somewhere that the paper is online as a pre-print.

Reviewer #2: This is an automated report for PONE-D-25-15324. This report was solicited by the PLOS One editorial team and provided by ScreenIT.

ScreenIT is an independent group of scientists developing automated tools that analyze academic papers. A set of automated tools screened your submitted manuscript and provided the report below. Each tool was created by your academic colleagues with the goal of helping authors. The tools look for factors that are important for transparency, rigor and reproducibility, and we hope that the report might help you to improve reporting in your manuscript. Within the report you will find links to more information about the items that the tools check. These links include helpful papers, websites, or videos that explain why the item is important. While our screening tools aim to improve and maintain quality standards they may, on occasion, miss nuances specific to your study type or flag something incorrectly. Each tool has limitations that are described on the ScreenIT website. The tools screen the main file for the paper; they are not able to screen supplements stored in separate files. Please note that the Academic Editor had access to these comments while making a decision on your manuscript. The Academic Editor may ask that issues flagged in this report be addressed. If you would like to provide feedback on the ScreenIT tool, please email the team at ScreenIt@bih-charite.de. If you have questions or concerns about the review process, please contact the PLOS One office at plosone@plos.org.

6. PLOS authors have the option to publish the peer review history of their article (what does this mean? ). If published, this will include your full peer review and any attached files.

**Do you want your identity to be public for this peer review?** For information about this choice, including consent withdrawal, please see our Privacy Policy .

Reviewer #1: No

Reviewer #2: No

---

## [Author Response · Author response to Decision Letter 1]

30 May 2025

Please see the "Response to Reviewers" file.

---

## [Decision Letter · Decision Letter 1]

16 Jul 2025

PONE-D-25-15324R1β-aminopropionitrile-induced thoracic aortopathy is refractory to cilostazol and sildenafil in micePLOS ONE

Dear Dr. Sawada,

Thank you for submitting your manuscript to PLOS ONE. After careful consideration, we feel that it has merit but does not fully meet PLOS ONE’s publication criteria as it currently stands. Therefore, we invite you to submit a revised version of the manuscript that addresses the points raised during the review process.

The authors will note that many details were requested by the Reviewers. As well, some questions will require new data or reanalysis of existing data. 

We look forward to receiving your revised manuscript.

Kind regards,

Jeffrey S Isenberg, MD, MPH

Academic Editor

PLOS ONE

Journal Requirements:

Additional Editor Comments:

The expert reviewers found positive aspects in the work. But there are limitations that mut be addressed. This will likely involve additional data and well as revision of the text. The authors are encouraged to consider each comment seriously.

Reviewers' comments:

Reviewer's Responses to Questions

**Comments to the Author**

1. If the authors have adequately addressed your comments raised in a previous round of review and you feel that this manuscript is now acceptable for publication, you may indicate that here to bypass the “Comments to the Author” section, enter your conflict of interest statement in the “Confidential to Editor” section, and submit your "Accept" recommendation.

Reviewer #3: (No Response)

2. Is the manuscript technically sound, and do the data support the conclusions?

Reviewer #3: Partly

3. Has the statistical analysis been performed appropriately and rigorously? 

Reviewer #3: Yes

4. Have the authors made all data underlying the findings in their manuscript fully available?

Reviewer #3: No

5. Is the manuscript presented in an intelligible fashion and written in standard English?

Reviewer #3: Yes

6. Review Comments to the Author

Reviewer #3: Comments:

The author did the research on two inhibitor drugs (phosphodiesterase inhibitors), one is cilostazol a PDE3 inhibitor and other is sildenafil a PDE5 inhibitor, to investigate the effect of these two drugs on the development and progression of disease thoracic aortopathy induced by β-aminopropionitrile (BAPN) in mice. Author also did bulk RNA sequencing to see the β-aminopropionitrile (BAPN) effect on Pde3a and Pde5a gene transcription level. Although the author found both genes were upregulated in the disease development but dietary administration of both drugs failed to alter the outcomes i.e. aneurysm formation, aortic rupture, and mortality. Finally author concluded based on the experimental data that cilostazol and sildenafil did not influence BAPN-induced thoracic aortopathy in mice.

The author has investigated a very important disease of the concerned. The study is well structured and addressed an important question. The most interested thing is the different effect of cilostazol and sildenafil on the disease. However, there are several major concerns must be addressed.

Major comments:

1) Author please explain the hypothesis of your study, is it only to find out the efficacy the inhibitors? As it is not clear with your experimental plan.

2) The author used a dose of 1 mg/day for each drug, how was this optimization done? Other studies often used 30-60 mg/day. The author should provide a dose optimization data as the dose author used for the study is considerably low and maybe that’s why lack of therapeutic effects observed. Pharmacokinetic data would be helpful.

3) Please justify why author chose 84 days of administration of treatments?

4) The BAPN concentration used in this study (0.5% wt/vol) which showed high mortality rates. There are previous study with low concentration (0.1-0.3%) as we know the BAPN effects are highly dose dependent. Author needs to justify dose choice based on their hypothesis or may be it is related to model also.

5) Why does the author only provide one time point mass spectrometry data? We know that the half-lives of cilostazol and sildenafil in plasma mice are relatively short (cilostazol: 2-4 hours, sildenafil: 0.3-2 hours).

6) Author did mass spectrometry which confirm the drug presence in plasma but did not address wheather the concentration of drugs were maintained throughout the experimental period or not as the half-lives of both drugs are relatively short

7) The author did bulk RNA sequencing, I don’t know how it will be helpful to author rather single cell sequencing will be helpful but I can assume the author has large data set but there is no evidence (eg. Bubble plot) mechanistic insight of drug effects or any molecular readouts in downstream pathways and author did not mentioned which cell types are involved in this mechanism. So the study lacks mechanistic investigation beyond RNA- sequencing data.

8) Author did not included any data on PDE enzyme activity , cAMP/cGMP levels or any signaling related study which limits its focus why drugs were ineffective?

9) The author states in the discussion section "These results indicate that BAPN-induced thoracic aortopathy is mediated by mechanisms distinct from other aortopathy models" by reinforcing this point with evidence or by emphasizing the ECM-dominant and non-inflammatory BAPN pathology that limits efficacy.

10) This study lacks histological evaluation, my suggestion is to include immunostaining data which would deepen the mechanistic insights.

11) The author should explain the novelty of this study in relation to other published studies as the present condition of the study only reflects the characterization of two drugs efficacy in thoracic region otherwise author should provide mechanistictic explanation for their negative findings.

Minor Comments:

1) In the abstract section line 29 author started with the describing about the disease but why “Phosphodiesterases (PDEs) regulate intracellular cyclic nucleotide concentrations” suddenly came please explain.

2) In the Introduction section line 46 “A key pathological feature of thoracic aortopathy and changes” Please check the grammer.

3) In line 53 author mentioned “This mouse model” which mouse model author wants to mention, please include that.

7. PLOS authors have the option to publish the peer review history of their article (what does this mean? ). If published, this will include your full peer review and any attached files.

**Do you want your identity to be public for this peer review?** For information about this choice, including consent withdrawal, please see our Privacy Policy .

Reviewer #3: No

---

## [Author Response · Author response to Decision Letter 2]

4 Aug 2025

Please see the Responses to Reviewers

---

## [Editor Report · Decision Letter 2]

12 Aug 2025

β-aminopropionitrile-induced thoracic aortopathy is refractory to cilostazol and sildenafil in mice

PONE-D-25-15324R2

Dear Dr. Sawada,

We’re pleased to inform you that your manuscript has been judged scientifically suitable for publication and will be formally accepted for publication once it meets all outstanding technical requirements.

Kind regards,

Jeffrey S Isenberg, MD, MPH

Academic Editor

PLOS ONE

Additional Editor Comments (optional):

The authors provided a second revised draft of their manuscript that addressed the comments put forward by the reviewer. Thier extra effort is appreciated.
---

## [Editor Report · Acceptance letter]

PONE-D-25-15324R2

PLOS ONE

Dear Dr. Sawada,

I'm pleased to inform you that your manuscript has been deemed suitable for publication in PLOS ONE. Congratulations! Your manuscript is now being handed over to our production team.

Kind regards,

on behalf of

Dr. Jeffrey S Isenberg

Academic Editor

PLOS ONE